# Advancements in the Design and Development of Dry Powder Inhalers and Potential Implications for Generic Development

**DOI:** 10.3390/pharmaceutics14112495

**Published:** 2022-11-17

**Authors:** Abhinav Ram Mohan, Qiang Wang, Sneha Dhapare, Elizabeth Bielski, Anubhav Kaviratna, Liangfeng Han, Susan Boc, Bryan Newman

**Affiliations:** Division of Therapeutic Performance I, Office of Research and Standards, Office of Generic Drugs, Center for Drug Evaluation and Research, U.S. Food and Drug Administration, Silver Spring, MD 20993, USA

**Keywords:** dry powder inhaler (DPI), orally inhaled drug products (OIDP), bioequivalence (BE), generics, advanced drug delivery

## Abstract

Dry powder inhalers (DPIs) are drug–device combination products where the complexity of the formulation, its interaction with the device, and input from users play important roles in the drug delivery. As the landscape of DPI products advances with new powder formulations and novel device designs, understanding how these advancements impact performance can aid in developing generics that are therapeutically equivalent to the reference listed drug (RLD) products. This review details the current understanding of the formulation and device related principles driving DPI performance, past and present research efforts to characterize these performance factors, and the implications that advances in formulation and device design may present for evaluating bioequivalence (BE) for generic development.

## 1. Introduction

The challenges surrounding the use of metered dose inhalers (MDIs) became a major driver for the development of dry powder formulations and dry powder inhaler (DPI) technologies [1,2]. However, the manufacture and development of these DPI products are rather complex and create challenges with respect to establishing bioequivalence (BE) between the proposed generic product and the reference listed drug (RLD), also commonly known as the brand name drug product. There are many different excipients and manufacturing methods that can be utilized to develop dry powder formulations, and each formulation requires careful study to understand how it affects product performance. Furthermore, unlike MDIs which generally follow a more standardized actuator design, there are many different DPI device types and designs. These DPI device factors also influences product performance as well as the ability for patients to utilize the device, since each DPI device can have a unique design and administration procedure. The aim of this review is to evaluate the current and future trends in DPI development and provide insight into formulation and device related principles driving DPI performance, past and present research efforts to characterize these performance factors, and the implications that advances in formulation and device design may present for evaluating BE for generic DPI product development.

## 2. Overview of DPI Products

DPIs are commonly used drug products for modern inhalation therapy of respiratory diseases that have specially designed delivery devices. These devices were developed to overcome certain limitations of MDIs, such as the complex coordination required for optimal lung delivery during times when patients may be undergoing bronchospasms [3], as well as the bulkiness and drug delivery duration of nebulizer devices. Most DPI devices are breath-actuated, which inherently avoids the need to synchronize device actuation with inspiration maneuvers.

Most DPIs contain three functional parts: powder drug formulation, drug dose measuring system, and a physical mechanism that allows dispersion of the powdered formulation. The following sections will examine each of these three aspects of a DPI, as well as the utility for patients and considerations for demonstrating BE.

### 2.1. Drug Formulation as a Powder

A DPI’s formulation is typically a complex dry powder mixture consisting of one or more active pharmaceutical ingredients (API) along with inactive, excipient ingredients. The use of excipients in the development of the formulation may not only enhance the chemical and physical stability of the API but can also impact the performance of the product. For example, this may include changing the dissolution of the API, the particle size, deposition to the lungs, and therapeutic efficacy. In general, excipients are usually “generally recognized as safe” (GRAS) substances that may improve the delivery and performance of a drug but cannot exert therapeutic effects by themselves [4]. Compared to other routes of administration, e.g., oral, topical, or parenteral, the number of excipients currently used or approved in inhalation drug products is fairly limited. The choice of excipients may depend on the disease being treated as well as the target region for delivery. Notably, the amount of the API in a single dose from a DPI is often very small, which makes dispensing a reproducible amount with each actuation challenging. Therefore, in most dry powder formulations, the excipients are used as carrier particles that provide bulk to the formulation, thereby improving the metering, dispensing, and handling of the formulation [5]. Traditional DPI formulations consist of micronized drug particles blended with lactose, a disaccharide composed of galactose and glucose, as coarse carrier particles. Lactose is an inexpensive excipient with an established stability and safety profile that is available in different grades [6,7]. Lactose is known to improve powder flowability thereby improving the reproducibility of the dose and as a diluent [8]. During drug formulation manufacture, lactose in the DPI formulation acts as a stabilizer for the spray drying process [8].

When selecting which excipient(s) to include in a DPI’s formulation, understanding how the excipient properties will influence the formulation or affect its biocompatibility with the site of action in the lungs is critical. Polymers, such as methylcellulose that are usually used in oral formulations, cannot be used as excipients in DPI formulations because these are non-degradable and, therefore, should not be delivered to the lungs. Similarly, polysorbates or oleic acid, used as excipients in MDI formulations, may not be used in DPI formulations due to their semi-solid or liquid states and low melting points which are not suitable for dry powder phase-based formulation. While lactose is a well-known DPI excipient, its powder structure may impact the inhalation efficiency of the DPI formulation. For example, amorphous lactose can give rise to strong particle–particle interactions that may impede aerosolization of the formulation [9]. To help overcome this, magnesium stearate (MgSt) can be employed as an adsorbed coating to reduce particle flocculation and thereby enhance the performance of lactose-based DPI formulations [10].

Sugars, such as mannitol and trehalose, are other excipients commonly used in DPI formulation development. Mannitol is a sugar alcohol and being a non-reducing compound, it is compatible with APIs containing amines [11]. Mannitol is also less hygroscopic than lactose and, therefore, cannot be used as a stabilizer for spray dried DPI formulations because of the rapid crystallization [12]. In addition, high amounts of mannitol can create a hyperosmolar environment and, therefore, are typically not used in DPI formulations for the management of asthma [13]. Trehalose is also a non-reducing disaccharide that may be used as a stabilizer for spray-dried inhalation powders. However, because of the hygroscopic nature of amorphous trehalose, it is often combined with other excipients, such as leucine [13].

Other prospective sugars have been evaluated, but they generally give rise to similar limitations as trehalose. For example, sorbitol, xylitol, and maltitol, have been evaluated as excipients for DPIs, but their hygroscopic nature and sensitivity to humidity have generally precluded their use in these drug products [14]. The use of amino acids such as trileucine as an excipient can improve powder dispersibility and reduce moisture uptake by the dry powder formulation. Trileucine is an efficient surface-active agent which provides a hydrophobic surface to the dry powder particles, thereby producing particles with low cohesivity, thus, improving the aerosol efficiency [15]. Fumaryl diketopiperazine (FDKP) is another excipient that is utilized with the proprietary TechnoSphere^®^ formulation, such as found in the recently approved Tyvaso^®^ DPI [16]. FDKP is a highly water-soluble compound that precipitates and agglomerates into low density particles under acidic conditions to form an encapsulating layer around the API [3].

### 2.2. Drug Delivery and the Device Landscape

There are many variations when it comes to DPI device designs, which are the result of optimization for formulation delivery and patient use. Some of the important functions of a DPI device are:Ability to protect the drug formulation from environmental factors (e.g., humidity, light, dust);Minimize residual drug remaining after device actuation;Consistently deliver a metered dose;Have a resistance appropriate to achieve the desired flow rate;Enable patient compliance and be easy to use with minimal dose administration steps.

Furthermore, depending on the frequency of dosage and the API, devices will be either a “single-dose” system or a “multi-dose” system, and can come as reusable or disposable devices. The formulation and clinical application also dictate how the drug will be stored in the device to maximize the emitted dose (ED) [17].

Currently, there are three types of DPI drug storage systems—capsules, blister packages, and reservoirs. Capsules for DPIs are typically composed of gelatin or hydroxypropyl methylcellulose (HPMC) [18,19], but may be composed of different materials depending on the formulation. It is desirable for the capsule composition to be inert and not interact with the formulation as this helps in maximizing the ED. These capsules are fitted into the DPI device and pin punctured so that upon inhalation, the drug is released from the capsule and delivered to the patient. In contrast, blister packages are usually foil-based containers that are either peeled or punctured, depending on the device mechanism, to release the drug powder formulation [20,21,22,23,24,25]. Finally, reservoirs are small container fittings for the DPI which carry the drug powder formulation as a bulk product and either go through a metering system for a multi-dose device or are stored in disposable reservoirs for single-dose devices. Below is a brief examination of some examples of different DPI device designs that are either marketed in the U.S. or elsewhere or are being researched. 

HandiHaler and Cyclohaler: The HandiHaler^®^ (Figure 1) and Cyclohaler^®^ DPI devices are classified as single-dose, capsule-based devices. Multiple drugs are delivered by the HandiHaler^®^ and Cyclohaler^®^ DPI devices—salmeterol xinafoate, beclomethasone dipropionate, ipratropium bromide, budesonide, formoterol fumarate, and tiotropium bromide. In the U.S., the HandiHaler^®^ DPI device is marketed to deliver tiotropium bromide under the brand name Spiriva HandiHaler^®^. This drug–device product has three major steps: after the capsule has been inserted into the device, it releases a single dose upon puncture on both long ends of the capsule, and the contained powder drug is delivered by breath actuation [3]. The HandiHaler^®^ has a slightly higher airflow resistance compared to the Cyclohaler^®^ [26]. Typically, it is understood that a higher airflow resistance indicates the need for a higher inspiratory flow rate, while a lower airflow resistance indicates the need for a lower inspiratory flow rate; however, this is not necessarily the case always for all devices that have a high, medium, or low airflow resistance.

Podhaler: The TOBI^®^ Podhaler^®^ (Figure 2) was approved in 2013 for the treatment of *Pseudomonas* infections in cystic fibrosis patients. The Podhaler^®^ (developed by Novartis Pharmaceuticals Corporation) was seen in a more positive light than nebulizers, which were inconvenient for transport and could easily be contaminated [3,28]. The TOBI^®^ Podhaler^®^ also utilizes a novel porous particle formulation technology known as PulmoSphere™ which is a registered trademark of Novartis AG, licensed to the Viatris Companies. PulmoSphere™ particles have advanced aerodynamic properties to enable lung deposition of the antibiotic tobramycin. The Podhaler^®^ device [28] is also used for the delivery of other antibiotics such as ciprofloxacin. It is a cylindrical shaped, capsule-based device which consists of a mouthpiece, chamber, body, and button to pierce the capsule so that the powder may be released upon breath actuation [3,28,29]. It is classified as a multi-unit dose device, and the formulation is phospholipid based. The Podhaler^®^ was designed to have a low airflow resistance to allow patients to generate high airflow rates (40–85 LPM) and attain reliable dose delivery [3].

The Ellipta^®^ is a family of DPIs utilized for the treatment of chronic obstructive pulmonary disease (COPD) symptoms and asthma (Figure 3). For example, Breo Ellipta^®^ contains double foil blister packs of fluticasone furoate and vilanterol trifenatate as maintenance therapy for COPD patients and in order to reduce exacerbations. The device comes pre-loaded with the drug blister packs and is actuated by releasing a lever on the side of the device and then inhaling through the mouthpiece. The device has medium resistance and is reported to have an intuitive device design for the patient [22,31]. 

The Dreamboat is a reusable multi-unit dose, cartridge-based device [3]. It is for the systemic delivery of insulin. The insulin powder is prepared by freeze-drying with a novel excipient, FDKP, to form porous particle TechnoSphere^®^. The pre-metered plastic cartridges contain a dose of either 4 IU or 8 IU insulin. Once the dose has been delivered, the cartridge is removed. The inhaler deagglomerates the powder in a convergence zone where two independent flow paths intersect. The inhaler has a relatively higher resistance; however, due to the population excluding those that have pulmonary diseases, the patient should be able to generate sufficient flow to actuate the inhaler.

The Twisthaler^®^ (Figure 4) is an example of a reservoir-based DPI for the delivery as part of Asmanex^®^ Twisthaler^®^ drug product that contains mometasone furoate indicated for the treatment of asthma. It is a reusable multi-dose, breath-actuated device [33,34]. Prior to inhalation, the patient must twist the mouthpiece until a click is heard to prepare the device with drug product for inhalation. The device is used for both children and adults and belongs to the class of higher-resistance devices [35].

The TwinCaps^®^ is a single-use disposable multi-unit dose inhaler, designed to be marketed as a pre-filled, low-cost inhaler to deliver large doses of drug. It was predominantly used in Japan for the systemic delivery of laninamivir (laninamivir octanoate hydrate), a novel neuraminidase inhibitor for the treatment and postexposure prophylaxis of influenza via pulmonary administration. The inhaler consists of two plastic parts: a plastic housing and a twin reservoir to carry the drug. The device has intermediate airflow resistance and is operated by sliding the reservoir chamber from side to side and inhaling. The device is primed by vertically tapping downward on a hard surface to make sure all the powder has settled to the bottom. The formulation is mixed with lactose blend for manufacturing the dry powder.

While each of these device designs comes with its benefits, it is also important to note that each design has its limitations. With capsule based DPIs that require single doses, for example, loading the capsule into the inhaler immediately before use may be a maneuver potentially inconvenient for some patients since this does not allow direct counting of the remaining doses [17]. Furthermore, properly loading the capsule-based DPIs requires a sequence of steps that may not be easy for patients with reduced dexterity. Some of these capsule-based DPIs require eight steps to inhale the medication (e.g., the Breezhaler^®^ and HandiHaler^®^), and this may not be as intuitive for patients and hence contribute to reduced compliance [17,18]. On the other hand, patients may have difficulty loading and cleaning the blister-based devices. If the blister packs are not completely pierced, it may lead to incomplete drug delivery, or patients may cover the air inlet holes with their mouths while inhaling due to incorrect positioning. With reservoir devices, the drawback may come from the patient not holding the device correctly in a way that the reservoir empties the full dose. The devices are summarized in Table 1 below. These considerations play a role into patient aspects of device usage, which will be discussed in the next section.

### 2.3. Patient Aspects

#### 2.3.1. Disease Conditions Treated

DPIs have been commonly used to treat patients with respiratory diseases, especially asthma and COPD. Critical to disease treatment and control is the delivery of the targeted dose from the device to the lungs. However, dose delivery is not only fulfilled by the optimal design of DPI devices but also affected by the inspiratory flow generated by the patient. This is largely determined by the severity of the patient’s disease. For example, asthma and COPD are both characterized by airway narrowing that limits air flow and gas exchange. Ultimately, the kinetic energy of the inspiratory air flow should efficiently deagglomerate dry powder and generate aerosol containing particles in the aerodynamic size range of 1–5 µm suitable for lung deposition.

Relevant to the inspiratory flow rate is the severity of disease. Here we will discuss the severity classification and treatment for both diseases which are closely related to the DPI performance. The Expert Panel Report 3 (EPR 3): Guidelines for the Diagnosis and Management of Asthma (2007) provides detailed clinical information on asthma management [37]. In general, disease severity can be determined using pulmonary function testing such as spirometry, which measures the air volume that is breathed in and out of the lungs in one forced breath. A common parameter measured in spirometry, FEV1, represents the forced expiratory volume of air in one second. A predicted FEV1 value is obtained from a given healthy population, taking into consideration other factors, such as age, gender, and height. The percentage of predicted FEV1 value in an individual patient can be used for severity classification. The severity classification of asthma and treatment in patients ≥ 12 years of age are shown in Table 2 [37]:

Similar clinical information in COPD patients has been discussed in the Global Initiative for Chronic Obstructive Lung Disease (GOLD) guideline (2022) [38]. The severity classification of COPD is shown in Table 3:

Asthma and COPD patients use a large variety of DPIs delivering different classes of medications for disease control. Although the treatment choice for COPD patients is typically individualized, based on a variety of factors including drug benefit/risk analysis and disease severity, these patients are often treated with long-acting muscarinic antagonist (LAMA) in combination with a long-acting beta_2_-antagonist (LABA). The LAMA-LABA dual treatment has been found to significantly improve symptoms and lung functions, reduce exacerbation rates, and decrease hospitalizations. Regardless, the severe impairment of lung function in severe asthma and COPD patients can pose a challenge to the targeted dose delivery for DPIs, as these patients may generate insufficient inspiratory flow for dose delivery. Moreover, device resistance is also a critical factor that affects the turbulent energy required for powder deagglomeration and dose delivery.

#### 2.3.2. Importance of Inhalation, User Interface, and Coordination

Patient-controlled factors, such as whether a patient’s training to use the DPI product is successful, impact a patient’s inhalation technique, adherence, and, ultimately, the success of treatment outcomes. DPIs, whether breath-actuated or passive systems, were developed to address challenges of inhalation and coordination that patients face when using powder MDIs and are generally considered to minimize the patient–device coordination [39,40,41]. Passive DPIs allow for the patient to coordinate the actuation process by using their own breath to initiate and complete dose delivery from the device. While this process eases the coordination between actuation and inhalation steps, the use of passive DPIs still require a sufficient inhalation technique to achieve the desired delivered dose to the target regions within the lung [42]. This is critical as the peak inhalation flow (PIF) achieved by patients through each DPI product has been shown to be tied to the clinical efficacy [43]. A proper inhalation technique from a patient does rely on successful patient training by a healthcare provider, which is recommended in clinical practice guidelines [44]. However, in the real word, both patients and physicians struggle to master the proper inhalation technique due to a myriad of factors. Some patient populations may struggle in understanding critical steps for DPI use, may be physically or cognitively limited due to age (most notably children and the elderly) or lack of education, or may receive insufficient training from their healthcare professional [34,37]. Healthcare professionals may also struggle with training by failing to understand the proper DPI device technique, lack of time for appropriate review of the patient’s inhalation technique, or due to poor communication with the patient [32,34,37]. The lack of confidence in the proper inhalation technique felt by a patient can lead to nonadherence and poor disease control exacerbations, hospitalizations, and, in some cases, the need for oral medications to control the patient’s asthma or COPD condition [34,37,45]. Thus, proper education, communication, and training regarding the patient and healthcare level are simple steps that are likely to improve therapeutic outcomes.

While proper education and training may address some concerns, other confounding factors that impact proper inhalation technique and clinical outcomes are related to the DPI device design and the variety of DPI device types that exist on the market. Passive DPIs utilize the patient’s inhalation as the energy source to achieve the ED from the device. Each DPI device design has a unique internal geometry, airflow path, and other internal features to assist in the deagglomeration process of the API from their carrier particles and achieve a desired aerosol performance [35]. Since each DPI design uses a unique dispersion process and/or internal geometry and airflow pathway, the internal resistance to airflow is also unique to each DPI device design [35,46]. The correct inhalation technique is clinically important because these devices utilize the interaction between the patient’s inhalation flow and the internal resistance of the inhaler to generate the turbulent airflow energy necessary to deagglomeration and aerosolize the powder formulation and achieve the desired respirable dose [36,47,48,49,50]. Thus, the patient’s inhalation effort and the device’s internal resistance to airflow are critical to achieve the desired lung dose.

When developing a generic DPI, an emphasis on the device design is expected as the DPI device and its interaction with the formulation and patient’s ability to properly use the device are essential to achieve an optimized aerosol performance [35]. Therefore, a generic device should be substitutable to the RLD DPI device when used by the patient. Specifically, a generic DPI is expected to contain similar external operating principles and external critical design attributes (e.g., size, shape, and operating steps) as the brand (RLD) product it intends to reference. The generic DPI device is also expected to contain the same metering principle to the RLD DPI device (e.g., a metered multi-dose format for reservoir or blister based DPIs, a pre-metered single-unit dose capsule-based format for capsule based DPIs). This ensures effective use of the generic DPI product when substituting for the RLD DPI product by minimizing patient confusion and ensuring ease of patient use [51]. From a patient-use perspective, it is desired that the proposed generic device has similar airflow resistance as the RLD DPI device. This helps guarantee the intended patient population is able to operate the device without significant change in inspiratory effort while achieving the same dose of medication [45,52].

The user interface encompasses the external critical design attributes, those aspects of the device related to how a patient uses the product for drug administration [46]. The user interface includes all components of a drug–device combination product that a user interacts with, including the delivery device, associated device controls and displays, product labeling, and product packaging. When developing a generic DPI product, the generic manufacturer is expected to assess the user interface of their proposed product as described in the draft FDA guidance for industry, *Comparative Analyses and Related Comparative Use Human Factors Studies for a Drug-Device Combination Product Submitted in an ANDA* [44]. This outlines the U.S. Food and Drug Administration’s (FDA) current thinking on what and how to compare the user interface of a proposed generic to its intended RLD product. As explained in this guidance, the assessment includes a labeling comparison, a comparative task analysis, and a physical comparison of the delivery device constituent part to analyze any potential differences (e.g., no design differences, minor design differences, or other design differences) between the user interface of the generic product and RLD product [44].

Overall, the design of the user interface for a generic DPI is not expected be identical to its RLD, but it is expected to produce the same clinical effect and safety profile as the RLD under the conditions specified in the labeling without the intervention of a health care provider and/or without additional training prior to use. Generally, any differences in user interfaces should be adequately analyzed and scientifically justified via a comparative (threshold) analyses and, if necessary, additional data such as a comparative human factors study. Considerations, such as the indication, context of use of the product (emergency vs. non-emergency use, daily vs. intermittent use), the end user (pediatric and/or adult patients vs. health care provider), potential use errors, and user error risks are considered during the user interface assessment of any generic and RLD DPIs. Thus, it is critical to consider these aspects when developing a generic drug–device combination product.

### 2.4. General BE Recommendations

To establish that a proposed generic drug product is bioequivalent to its RLD, the generic drug applicant must demonstrate that there is no significant difference in the rate and extent of the API (in pharmaceutical equivalents) becoming available at the site of action as the reference standard when the same molar dose is administered in an adequately designed study. As detailed above, the performance of inhalation drug products like DPIs can be influenced by many different factors, such as the various aspects of the formulation, its interaction with the device characteristics, and the disease condition being treated. With such a varied number of influencing factors, DPIs are considered complex drug–device combination products that require a multi-faceted approach to demonstrate that a generic company’s DPI is bioequivalent to the RLD. The FDA has generally referred to this approach as the agglomeration weight of evidence approach as it includes a combination of in vitro and in vivo studies to evaluate whether differences exist in product performance, systemic exposure, and local drug delivery between a generic product and its RLD [53]. The FDA has published product-specific guidances (PSGs) that outline the FDA’s recommendations for demonstrating BE for a number of DPI products [54].

For DPIs, the evaluation of product performance is conducted through in vitro BE studies that measure single actuation content (SAC) and aerodynamic particle size distribution (APSD). In general, in vitro BE studies are sensitive to detecting differences in formulation, device, and manufacturing process that could impact product performance. For DPI products, both SAC and APSD studies have been recommended as they are believed to be relevant to the lung regional and total drug deposition performance for these products. Importantly, these studies are also conducted across a range of airflow rates that encompass the inspiratory flow ranges of the indicated patient population, thereby ensuring that any variability in a patient’s ability to inhale through the proposed generic DPI will not result in performance differences that may affect its BE to the RLD.

The in vivo BE studies typically recommended for DPIs include both pharmacokinetic (PK) studies and comparative clinical endpoint or pharmacodynamic (PD) studies. PK BE studies are included in the weight of evidence approach to evaluate whether there is a difference in the systemic exposure of the generic product and RLD that may lead to differences in side effects or any adverse reactions. It is recommended that these studies use a single-dose design in healthy volunteers instead of patients, given that healthy volunteers generally provide a lower PK variability since patient-related factors, such as disease severity, are not present. As with the in vitro BE studies, PK BE studies for DPIs are recommended for each strength, using a dose that, while based on the analytical assay sensitivity, requires a minimum number of inhalations. For establishing BE in local drug delivery, comparative clinical endpoints or PD BE studies are recommended to be conducted in one of the indicated patient populations using the lowest dose listed in the approved drug label. Comparative clinical endpoints or PD BE studies are often considered the most challenging studies for generic applicants to complete since these studies are generally considered to be less sensitive in detecting formulation difference as compared to other methods. Their use of the more variable patient population as well as the challenges stemming from the available endpoints for measuring local treatment effects often requires significant numbers of patients in each treatment arm to achieve sufficient statistical power for evaluating BE. However, while these challenges often result in a study design that is more costly and with a longer study duration that other BE studies, comparative clinical endpoint or PD BE studies are still included in the weight of evidence approach as they serve a confirmatory role for establishing equivalence in local drug delivery. With that said, identifying alternative approaches to comparative clinical endpoints or PD BE studies remains an active area of research and interest for the FDA.

In addition to the in vitro and in vivo BE studies described above, the FDA generally recommends that a generic DPI product be formulated qualitatively (Q1) and quantitatively (Q2) in the same way as the RLD product and have device similarity, as detailed earlier in this review. With that said, it is also important to note that, as per the U.S. Code of Federal Regulations (CFR), DPI generic products are not required to be formulated the same as their respective RLD product. Therefore, the FDA’s BE recommendations include approaches for generic DPI formulations with greater than 5% differences in an excipient amount compared to the RLD, so long as the generic applicant can provide adequate justification that this difference does not affect the product’s safety or efficacy. This may include in vitro testing across multiple drug-to-excipient ratios. In general, inclusion of formulation sameness criteria as part of a BE recommendation reduces the likelihood that a generic product will not be bioequivalent to the RLD. In addition, the criteria reduce the chance that differences in the excipients may lead to changes in safety of the product.

## 3. Understanding of the Advancements in Formulation and Aerosolization Factors and Regulatory Considerations

### 3.1. Understanding of Formulation and Aerosolization Factors

#### 3.1.1. Relationship between Formulation and Device

A majority of DPI formulations consist of micronized API blended with larger carrier particles that enhance flow, reduce agglomeration, and aid in dispersion. These carrier-based DPIs are manufactured using a process in which an API is incorporated into a drug product, and the formulation development has been challenging due to the molecules with pharmacologic activity often having poor physicochemical properties. The complex relationship between physicochemical properties of the DPI formulation and the design of the device ultimately impacts the fluidization, deagglomeration, and aerodynamic particle size distribution of the drug particles [55], which in turn, influence the local deposition of the drug in the lungs [56]. An understanding of the physical properties of DPI formulations, especially their size, morphology/shape, and surface roughness, is important since they impact dry powder delivery. Aerosol delivery from DPIs involves powder fluidization and powder deagglomeration. The success of these two phenomena to occur depend on the cohesive forces (i.e., electrostatic, capillary, van der Waals) among dry powder particles that must be overcome by forces generated by the airflow through a DPI [57].

As mentioned earlier, excipients used in the DPI drug formulations have a significant effect on both the product performance and local safety [4]. The physicochemical properties of the excipients, such as size and morphology, have a significant effect on the performance of the DPI [58,59,60,61,62]. Given that the number of excipients currently approved for pulmonary drug delivery is limited, increased drug delivery efficiency could be achieved by developing optimized powder formulations with carefully chosen excipients. The main goal of this strategy is to incorporate desirable attributes for the drug particles, including improving dispersibility, adjusting particle size distribution, enhancing drug stability, optimized bioavailability, sustained release, and precise targeting into the formulation.

Determination of the API polymorphic form is a critical part of the drug product formulation development process as different API polymorphs have different properties, such as solubility, stability, and even bioavailability, as they are at distinct energy states [63]. Moreover, polymorphs often differ in density, melting point, and hygroscopicity. The most stable polymorphs are typically chosen to reduce the transformation risk during the processing and storage for formulation development. Particle size is another important design variable of a DPI formulation, and the size distribution is normally bimodal because the formulation contains micronized API and larger excipient carrier particles. Since the surface morphology contributes to surface area of particles, particle morphology can also be exploited for DPI formulation design [64,65,66]. The micronized particle normally will have very high surface area and energy after high-energy separation of the device, which will result in poor flow and a higher tendency to agglomeration. Ideally, the forces between the contact area should provide enough adhesion between drug and carrier for a stable formulation, while allowing easy separation upon inhalation. Powder deagglomeration is also believed to occur via impaction; the magnitude of impact-based events may exceed those of flow-based events and could potentially be the dominating factor in powder deagglomeration [46]. In addition, many critical DPI particle properties, such as size, packing, shape, surface topology, crystallinity, surface energy, moisture content, and triboelectric charging, depend on the processing history of the powder.

Several research studies highlighting the intricate relationship between aerosolization performance of a DPI and factors related to the device and formulation have been conducted [26,67,68]. FDA has also conducted research (Contract HHSF223200910017C) to identify critical formulation attributes of two different DPI devices that have the same dose metering principle and similar device resistance via a quality by design approach. A central question for this research was if equivalent in vitro performance be achieved by adjusting the identified critical formulation attributes. The first study of this project assessed if the two DPI devices could generate comparable aerosolization performance under three flow rates. Both the theoretical and experimental results suggested that only matching the device resistance between DPIs may not be sufficient to obtain comparable in vitro performance. The second study evaluated the effect of critical device and formulation attributes on the comparability in aerosolization performance. The results showed that, with adequate and systematic modifications of key critical device attributes (such as the use of airflow channels to alter achieved pressure drop and particle velocity within the device), the in vitro comparability could be achieved. The last study focused on assessing the effect of formulation critical attributes (e.g., API particle size, presence/absence of lactose fines, and degree of API:excipient adhesion) on the DPI aerosolization performance [61]. The outcomes from this study demonstrate the importance of understanding how critical device and formulation attributes affect the DPIs aerosolization performance, either when developing a new DPI or optimizing a potential generic DPI to achieve BE.

#### 3.1.2. Novel Formulations and Manufacturing Techniques

In case of carrier-based DPI formulations, the micronized drug particles are aerosolized and separated from the carrier (from drug–carrier mixtures) or deagglomerated and delivered into the patient’s lung. Although easier to formulate and adaptable to different particle size distributions, these carrier-based DPI formulations suffer from low blend uniformity. Another challenge with carrier-based formulations is the inability to meet pulmonary drug payload requirements; this is especially evident with anti-microbial drugs that require a high drug payload in a single inhalation to achieve the target lung concentrations [69]. To enhance the dose uniformity and improve high dose delivery of dry powder formulations to the lungs, recent research has focused on developing engineered powder formulations in combination with new DPI devices. These engineered particles are formulated to achieve high fine particle fraction by lowering particle size (e.g., nanoparticles), or by influencing particle density (e.g., porous particles) to reduce the aerodynamic diameter and enable alveolar delivery [70,71,72,73,74].

Improved aerosolization can also be achieved by modifying the shape, surface composition, and roughness, and introducing a surface charge to the particles that lowers the inter-particulate forces by increasing interparticle separation and lowering the contact area [75,76]. These engineered particles are prepared using traditional techniques, such as milling and crystallization, as well as with more advanced technologies, e.g., spray drying, spray freeze drying, or supercritical fluid technology [77]. These approaches can be classified into top-down (e.g., jet milling) and bottom-up (e.g., spray drying) processes depending on the workflow process [78].

A key parameter of dry powder aerosol performance is the powder density. This is a direct result of formulation and manufacturing processes. Most available DPIs have been made with particle mass densities of 0.5–1 g/cm and mean geometric diameters < 5 µm to avoid excessive deposition in the device and the oropharyngeal cavity. Research in the late 1990s showed that low density, very light particles with mass densities of 0.4 g/cm and diameters > 5 µm can aerosolize from a DPI more efficiently due to their large size and low mass density. The highly porous nature of these particles combined with their larger diameters resulted in similar aerodynamic performance as smaller, nonporous particles. Increased aerosolization efficiency of large, light particles combined with improved efficiency to enter the lungs lowers the probability of deposition losses, thereby increasing the systemic bioavailability of an inhaled drug. Moreover, reduced phagocytic clearance for such particles was shown to maintain a longer residence time [79]. Such porous particles with high porosity and low tapped density can be prepared using processing methods like spray drying, which offer greater control of particle properties, including particle size and distribution, morphology, porosity, density, and surface energy.

Porous particles with geometric sizes in the range of 5–30 μm (large porous particles) have been used in INBRIJA^®^, levodopa inhalation powder, that uses a capsule-based inhaler. It is a low-density powder formulation in which the biocompatible excipients (poly(lactic acid-co-glycolic acid) [PLGA]) form an aerodynamic structure that has shown improvements in aerosolization and phagocytic clearance from the lungs. The initial studies in rat models revealed that levodopa from porous particles was delivered to the alveolar membranes to be rapidly absorbed to the pulmonary capillary network and, thus, into the arterial circulation and to the brain, offering a more rapid and robust delivery [80,81].

Various methods have been used to manufacture porous particle formulations. Emulsion-based spray drying is used in the manufacture of PulmoSphere™ particles to obtain phospholipid porous particles with a sponge-like morphology for high-dose delivery of tobramycin (TOBI Podhaler). The process of spray drying to achieve the specific particle morphology of tobramycin inhalation powder is described in Figure 5. The emulsion is formulated using phospholipid (distearoylphosphatidylcholine [DSPC]) as a dispersion stabilizing agent, a surface modifier such as calcium chloride (CaCl_2_), and a pore forming agent such as perfluorooctyl bromide (PFOB). The pore forming agent is almost completely evaporated from the final powder during the drying process. The drug in the phospholipid matrix of the particles obtained by emulsion-based spray drying may be amorphous in nature, which could pose challenges to the long-term physicochemical stability of these powders. To maintain crystallinity of API(s) with porous particle formulations, suspension-based spray drying has been used in preparing porous particles to minimize drug particle dissolution in the liquid feed [82]. A number of studies in the literature have reported using the PulmoSphere™ technology for different drug molecules: human immunoglobulin (hIgG), gentamicin sulphate, salbutamol sulphate and budesonide for DPIs and MDIs [74,83,84,85,86].

Powder formulations containing PulmoSol™ technology used in EXUBERA^®^ (developed by Nektar Therapeutics) or freeze drying (e.g., TechnoSphere^®^ technology used in AFREZZA^®^ and TYVASO^®^ developed by MannKind Corporation). In these cases, insulin is preserved chemically and physically in the form of a powder that presents good dispersibility and aerodynamic properties suitable for pulmonary delivery to provide pulmonary bioavailability of 8–25%. Other types of engineered particles that are in early phase of development include liposomal dry powder formulations, where drug-encapsulated liposomes are converted into a dry powder form, particles coated with lipids, polymers, or leucine, and excipient-free nanoporous/nanoparticulate microparticles [8,88,89].

Research on porous particle preparation methods and their performance are summarized in Table 4. Currently, the methods of manufacturing porous particles can be categorized as either “non-freezing induced” (e.g., spray drying, supercritical fluid technologies) or “freezing induced” (e.g., spray freeze drying) through powder technology [90].

Spray-drying technology is a developed technique that involves atomizing the drug formulation solution into liquid aerosol particles under a drying gas stream contact. The dry solid particles are produced through the evaporation process in the spray drying chamber and can be collected using a cyclonic powder collector or electrode separator [97]. The physicochemical properties (i.e., surface area, size, and shape) of the generated solid particles could be adjusted by tuning the processing parameters of spray drying, which include temperature, feed pressure, feed rate of the drug formulation solution, air flow rate, and nozzle type [98].

Supercritical fluids (SCF) have been used for the formulation of DPIs as a cost-effective and non-toxic method to modify the solid-state form of the dry powder particles [99,100]. The advantages of the SCF are that it can dramatically reduce the use of organic solvents in the manufacturing process, increase the ability to modify the solid state of the powder produced, and tune the conditions to give rise to a desired particle size and size distribution. Supercritical CO_2_ is the most commonly used SCF technology in FDA approved DPI products, as it is recognized as safe and non-combustible. Supercritical CO_2_ can form the porous matrices due to its low viscosity, high diffusivity, and null surface tension [94,97]. With the affinity and high solvation power of supercritical CO_2_, the precipitation of compressed CO_2_ antisolvent (PCA) and supercritical fluid antisolvent techniques provide the advantage to achieve the desired size for inhaled particles [101]. A good example is insulin-loaded poly-L-lactide porous microspheres which is produced by PCA method and ammonium bicarbonate as porogen [94].

Recently, aerogels (which can be manufactured using SCF technology) have been explored for use in porous particle formulation development due to their diversity of textural properties and porosity [102]. The high porosity of aerogel particles provides a way to tailor the aerodynamic behavior and dispersibility, potentially providing a way for lung delivery with particles whose size would normally preclude them from entering the lungs. Pulmonary drug formulations with aerogel carriers have a low flow rate dependence on the respiratory capacity of the patient when used with DPIs. In addition, the aerogel’s high surface area can dramatically improve the drug dissolution rate. Given these characteristics, aerogels may have great potential in the development of novel formulations for DPIs [80].

Spray freeze drying (SFD) is another well-developed technique for large porous particle (LPP) generation in the inhaled drug field. It has a high production rate and is suitable for thermally sensitive materials. The whole process consists of three sections which includes atomization, freezing, and lyophilization. The produced LPP have a desired aerodynamic deposition and improved solubility of the formulations. SFD has been successfully applied to generate voriconazole-loaded LPP for the treatment of pulmonary aspergillosis. Published reports have shown that dissolution studies with voriconazole can be rapidly released from the LPP based formulation, as compared with its release from solid voriconazole particles that required 2 h to dissolve [96].

### 3.2. Methods for Characterizing DPI Formulation Properties and Performance

DPI microstructure including particle size, particle morphology, surface roughness, and interfacial chemistry have been shown to play a key role in governing particle–particle interactions. These properties also govern the relationship between the DPI device and the deagglomeration efficiency of the powder formulation [67]. As such, numerous methods have been identified that can characterize these properties, which can be beneficial when attempting to optimize a DPI’s formulation and performance. Particle morphology can be qualitatively assessed using scanning electron microscopy (SEM). Particle micromeritics and physical structure (e.g., specific surface area, porosity, and pore volume) can be quantitatively evaluated using gas adsorption (e.g., Brunauer, Emment and Teller (BET) specific surface area) measurements and mercury porosimetry. Differential scanning calorimetry (DSC) and X-ray powder diffraction (XRPD) can be used to assess the solid-state morphic form of the API and excipients in the formulated DPI product. The drug−drug and drug−carrier inter-particulate forces can provide information on the ability of the drug particles to be detached and inhaled by the patient using the device. Surface energy of powders as determined by inverse gas chromatography (IGC) has been correlated to powder dispersion [103] and may play a role in batch variability in DPI formulations [104]. Cohesive–adhesive balances (CAB) determined via colloid probe measurements using atomic force microscopy (AFM) can help understand and optimize the characteristics of carrier-based DPI formulations through quantification of the balance of inter-particulate forces of the API within DPI systems [105]. For protein or peptide containing powders, secondary structure changes can be evaluated using circular dichroism (CD) spectroscopy, while tertiary structure changes can be evaluated using protein melting endotherm, and agglomeration can be analyzed using dynamic light scattering (DLS) and size exclusion chromatography coupled to high performance liquid chromatography (SEC-HPLC) [88].

In addition to particle size and morphology, surface composition also influences cohesive forces and, ultimately, aerosol performance. X-ray photoelectron spectroscopy (XPS) as well as confocal Raman can be used for assessing the surface chemistry of powders [106,107]. Recently, novel characterization techniques combining optical photothermal infrared (O-PTIR) spectroscopy and atomic force microscopy infrared (AFM-IR) spectroscopy have been developed to resolve the distribution of excipient and drug particles at a submicrometer and nanometer scale. These techniques have been shown to successfully map the drug distribution in individual aerosol particles, such as spray-dried powders and lactose-based DPIs [108]. Morphologically–directed Raman spectroscopy (MDRS) which combines particle imaging with Raman spectroscopy provides a single integrated platform for structural analysis characterization of particle size and shape, as well as chemical determination such as the polymorphic form of APIs [109]. Interestingly, research studies funded by FDA through a contract with the University of Bath (Contract HHSF223201710116C) have suggested that techniques such as MDRS that can distinguish the various drug–drug and drug–carrier agglomerates within a given formulation may be able to inform on potential differences in DPI performance. In these studies, in vitro dissolution performance between different DPIs containing fluticasone propionate and salmeterol xinafoate was found to relate to the types and amounts of particular agglomerates within a given formulation [110]. This and other associated studies suggest that MDRS combined with dissolution may provide valuable insights into the microstructure of the aerosolized dose of DPIs [111].

Translating properties of single particle (or sampled particles) obtained from SEM, AFM and MDRS to the bulk powder, particularly inside the capsule or blister, is challenging. In such cases, powder analytical techniques involving penetrating radiations such as X-ray microscopy (XRM) or X-ray computed tomography (XCT) could be employed for non-destructive analysis of the bulk powder to obtain size and shape (sphericity/aspect ratio) distributions [112]. Although other imaging techniques such as SEM may offer higher resolution, an advantage of XCT is that it allows the powder to be examined from any desired angle with the minimum sample preparation to determine the initial particle matrix inside the blister or capsule. The combination of XCT and simulations has been used for understanding how the initial particle orientation and packing can affect aerosolization behavior [105]. However, an additional consideration when performing these single or small sample size analysis methods is the time required to complete the measurement. In contrast, laser diffraction is a lower resolution but more widely available technique that can provide faster analysis of particle size distributions of the DPI, as compared to the more complex techniques discussed above that provide better resolution.

In terms of in vivo based DPI performance characterization methods, recent FDA research studies conducted by University of Florida (Contracts HHSF223201110117A, HHSF223201000090C, and HHSF223201610099C) have explored whether systemic PK measurements can detect differences in regional lung deposition of DPI drug products [113]. Three carrier-based dry DPI formulations of fluticasone propionate were developed to achieve deposition in various regions of the lung (i.e., central vs. peripheral) yet providing similar lung doses (i.e., similar amounts of drug depositing in the lung). The formulations were prepared using the same batch of API to achieve similar dissolution rates for all three formulations. The PK studied utilized a randomized, double-blind, four-way crossover design in 24 healthy volunteers. Results suggested that PK may be capable of distinguishing between DPI formulations with apparent different regional depositions, albeit the sensitivity may vary between PK parameters, such as Cmax and AUC, and be influenced by how dose normalization is conducted in the study. Furthermore, PK parameter variability could present challenges to using this approach in a regulatory setting if this variability limits the ability to correlate systemic exposure and regional deposition.

Finally, modeling and simulation techniques can be advantageous in identifying relationships between the formulation and device properties of a DPI and its performance. For example, computational fluid dynamics (CFD) can be used to understand the air flow and deagglomeration in the inhaler device by tracking single particle motion in the device. On the other hand, information about agglomerate break-up during impaction and interaction with the air turbulence may require modelling techniques such as the discrete element method (DEM) [103,114]. The complexity of aerosol delivery through a DPI and the deposition in the respiratory airways emphasize the critical role of model validation with in vitro and in vivo data for a thorough and reliable analysis [115]. Complete understanding of a dry powder aerosol delivery system will likely continue to involve advanced analytical techniques that are coupled with modeling and simulations [116], thus highlighting the importance of understanding the complex nature of both the powder and its interaction with the device before and after aerosolization.

## 4. Advancements in DPI Device Technologies

In addition to the advancements in DPI formulation design, advancements are also continuing on the device front with the introduction of new designs and digital technologies. With the inclusion of digital technology, these digital DPI platforms have the ability to provide additional information on inhaler performance and/or use that may minimize device use errors. This has shown a positive impact on product use and performance, as well as patient satisfaction, compliance, and adherence, thus, overall clinical outcomes [117]. A recent review by Xiroudaki et al. 2021 summarizes the current landscape of digital DPIs [109].

The currently FDA-approved digital DPIs include the Digihaler^®^ used with ProAir^®^, AirDuo^®^, and Armonair^®^ products, which were approved in 2018, 2019, and 2020, respectively [109]. The Digihaler^®^ contains an integrated digital sensor encompassed in an electronic module (eModule) on the top portion of the inhaler that can record usage (date and time), inhalation data relevant to the inhalation technique (e.g., PIF and flow volume), and provide medication reminders [109,118]. The accuracy, benefits, and outcomes of such digital DPIs, such as the Digihaler^®^, are beginning to emerge [110,119,120]. Other FDA-approved digital DPIs included add-on sensors rather than being fully integrated in the DPI device. The Propeller^®^ sensors designed for several DPI devices including Diskus^®^ (GlaxoSmithKline, Brentford, UK), Ellipta (GlaxoSmithKline, Brentford, UK), and Neohaler (Novartis, Basel, Switzerland) were approved by the FDA through the 510(k) pathway in 2015, 2016, and 2018, respectively [109]. The Propeller^®^ sensors technology can record and monitor actuation activity (date and time) and utilizes Global Positioning System (GPS) technology to identify environmental asthma exacerbation triggers. Additionally, the Hailie^®^ sensor (approved via the 510(k) pathway) was developed for MDIs as well as the Diskus^®^ (SmartDisk™) and HandiHaler^®^ (SmartHandy™), which is capable of being attached to the inhaler to record and track medication usage and set reminders.

As the inclusion of digital sensors to DPIs (either added on or fully integrated) become more commonplace, the real-world evidence of their potential benefits may begin to show, such as changes in patient adherence and clinical outcomes. As mentioned earlier in this review, other inhalation products such as MDIs and nebulizer-based products can present challenges to patient adherence and outcomes due to their need for better coordination with the patient, or the potential device bulkiness and long duration of administration. While DPIs can address these challenges due to their passive drug delivery approach and size, the cost for patients to use a DPI over other inhalation products may also be a consideration that impacts patient adherence and clinical outcomes. So, while the addition of digital versions of DPIs may offer benefits to patients, these would need to be considered along with any differences in costs for the patient when attempting to understand potential changes to patient behavior such as adherence to a prescribed treatment. Currently, there are no approved generic DPIs that incorporate a digital feature. These digital technologies present a new challenge for generic DPI development given the uncertainty with degree to which a generic DPI will be required to include digital technologies, or record and/or communicate this information to the patient, in order to match the RLD DPI. With that said, generic competition with these DPIs is expected to lead to reduced costs for the patient, which has been reported following the introduction of the first Advair Diskus generic [121]. As this new area continues to develop and bring with it new regulatory challenges, the FDA continues to evaluate these challenges through regulatory science initiatives that will aid future generic development.

Lastly, the introduction of the Staccato inhalation platform provides an example of the continued device design advancements for inhalation powders. The Staccato device stands out among other inhalation devices due to its drug delivery mechanism. The device uses breath actuation, where an inhalation sensor triggers a heating element on the device that heats up a thin film of the formulation, causing it to sublimate and aerosolize the dose as dry particles [122]. Notably, the Staccato device’s use of a thin film formulation makes it distinct from other carrier-based DPI formulations as the formulation is not in a powder form prior to actuation. Currently, the Staccato device is used in the FDA-approved product Adasuve (loxapine) inhalation powder.

## 5. Additional Considerations for Bioequivalence

As detailed earlier, the FDA has generally recommended an agglomeration weight of evidence approach to establish BE of a proposed generic DPI to its RLD. These recommendations are product-specific and include tests to address the complexities of these drug–device combination products and the challenges they present for evaluating equivalence. The advancements in DPI formulation and device design may necessitate new experimental techniques to sensitively detect differences between proposed generic and RLD DPI products that could impact performance.

For example, study results on the role of the different types and amounts of agglomerates in a DPI formulation on its performance support the role that particle microstructure may play for these products. While the drug particle size is a well-known factor influencing regional deposition and subsequent drug release, the associated excipients in the depositing agglomerates may alter the surface area available for drug release of the API particles by preventing larger drug-drug agglomerates from forming. The speed at which these associated API particles become available for drug release may depend on the solubility of the excipient(s) in the agglomerates. Unlike lactose, which as a carrier particle has had its physical properties well studied, the various other excipient carrier particles (e.g., mannitol, trehalose) or surface modifiers (e.g., MgSt, trileucine) have limited information available for their potential influence on API drug release once formed into agglomerates.

Moving away from the more well-studied carrier-based DPI formulations, the advancements in particle engineering have identified novel excipients, such as Bis-3,6(4-fumarylaminobutyl)-2,5-diketopiperazine (FDKP) and DSPC, as well as advanced manufacturing processes, such as those listed in Table 3, which yield unique formulation particles with complex API and excipient associations. For example, porous particle formulations (e.g., PulmoSphere™ Technology) exhibit several unique performance features, such as similar aerodynamic characteristics to smaller particles, improved dose delivery, deeper regional deposition, and longer residence time, which stems from their high porosity and lower density. Rather than just simple adherence to an excipient particle surface, the API can be contained within these porous particles, which may present a complex relationship between their microstructure (e.g., shape, porosity, pore size, agglomerative potential) and the formulation’s performance, such as its dissolution rate. The multitude of porous particle manufacturing methods also raises questions on how differences in manufacturing method or process parameters affect particle microstructure and subsequent performance.

With these novel formulation-based complexities, additional considerations for microstructural differences and their potential impacts on DPI performance may warrant additional characterization studies to support product quality and demonstration of BE. For microstructural characterization, this review has described a wide assortment of analytical approaches that DPI formulators may use, including ways for evaluating small sample or single particle/agglomerate associations and morphology (e.g., MDRS, SEM, O-PTIR/AFM-IR, XRM/XCT, gas adsorption/mercury porosimetry), polymorphism (e.g., DSC, XRPD), and inter-particle interactions (e.g., IGC, AFM). However, additional research is still needed to determine which of these methods can most sensitively discriminate between a proposed generic DPI and its RLD when clinically meaningful differences in performance are present that would affect BE.

In relating the microstructural characteristics of a given DPI formulation to its performance, there are opportunities to utilize both in vitro and in vivo studies. For example, APSD studies using anatomical mouth-throat models and realistic breathing profiles, along with dissolution studies, are two potential in vitro approaches that may be useful for providing insight into how microstructural differences between DPI formulations may impact dose delivery, deposition, and drug release, provided discriminatory methods are developed and validated. For in vivo studies, PK studies may also provide a way to characterize microstructural impacts if differences are expected to affect regional deposition; however, more research is needed to better establish this method’s sensitivity to potential differences in the drug product microstructure. Lastly, given the various challenges and complexities presented by the different DPI formulations discussed in this review, the characterization methods selected to evaluate a proposed generic DPI and its RLD should be suitable to address the specific microstructural characteristics present in the DPI formulation. Therefore, characterization of DPI microstructure is likely to be product-specific and, so, should be appropriately justified.

When considering the implications of new digital technologies and device designs for BE, there is considerably more uncertainty at the present time with what potential differences that may be permissible for a generic DPI. As a starting point, and to facilitate generic drug–device development, the FDA has provided its current thinking in published guidance on considerations and approaches for evaluating substitutability of a generic drug–device product. Given the ever-changing landscape for device designs and user interfaces, FDA encourages prospective generic applicants to utilize available communication mechanisms, such as the pre-abbreviated new drug application (pre-ANDA) meeting request process [123], to obtain the Agency’s current thinking when considering their DPI device design and development program.

## 6. Conclusions

This review has provided a discussion of the FDA’s current understanding of the principles and contributions that a DPI’s formulation and device provides to its performance, as well as a brief overview of performance characterization methods that can be used during drug development and establishing BE. With the challenges that novel DPI formulations, manufacturing approaches, and device designs can present for evaluating potential generic DPIs, the FDA has funded research initiatives to better understand what factors are critical for ensuring BE with the RLD. This research has highlighted the role formulation microstructure can play in a DPI’s performance and identified novel in vitro and in vivo characterization methods that may be able to sensitively evaluate how microstructural differences impact performance. As DPI technology continues to advance, the FDA remains committed to building on the current understanding of DPI performance and encourages prospective generic applicants to communicate early in their development programs when challenges with evaluating DPI performance are encountered.

## Figures and Tables

**Figure 1 pharmaceutics-14-02495-f001:**
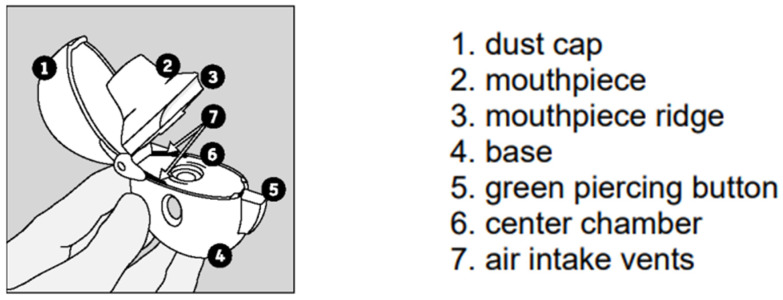
Diagram of the Spiriva HandiHaler [27].

**Figure 2 pharmaceutics-14-02495-f002:**
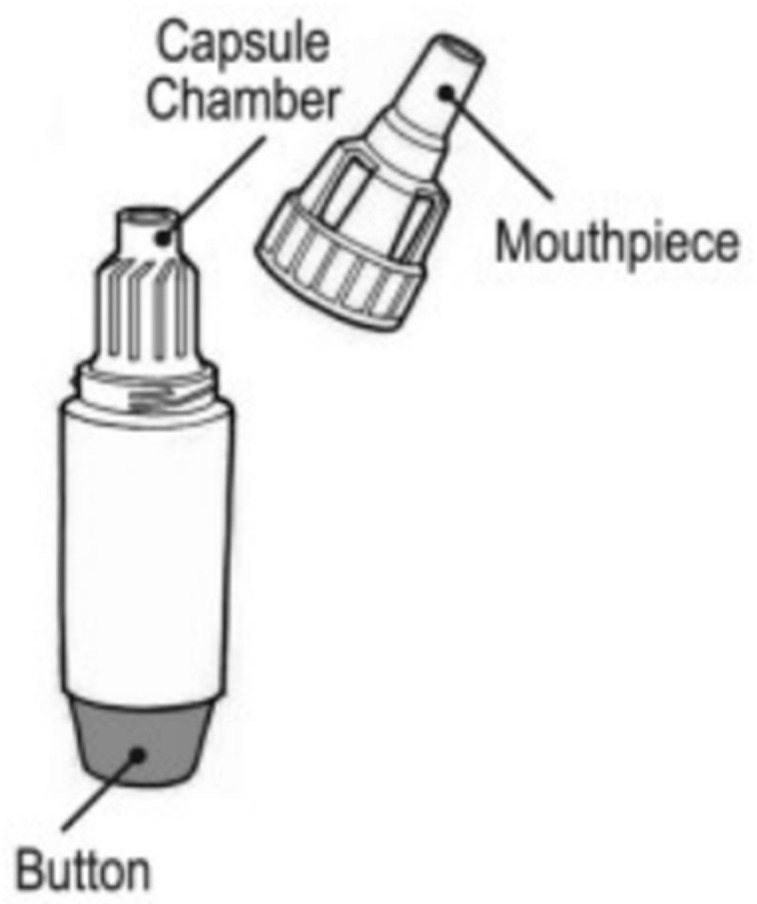
Diagram of the TOBI^®^ Podhaler^®^ [30].

**Figure 3 pharmaceutics-14-02495-f003:**
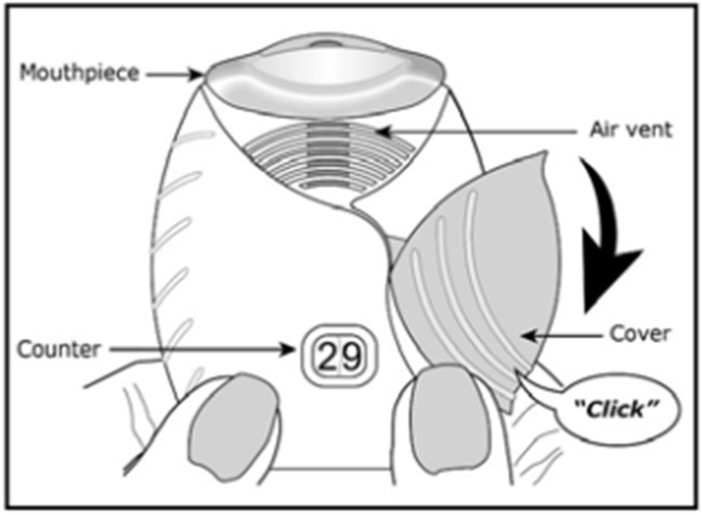
Diagram of Breo Ellipta^®^ [32].

**Figure 4 pharmaceutics-14-02495-f004:**
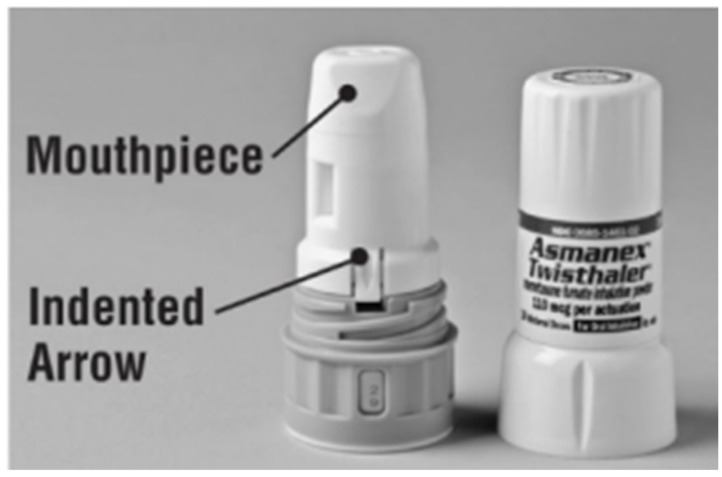
Asmanex^®^ Twisthaler^®^ [36].

**Figure 5 pharmaceutics-14-02495-f005:**
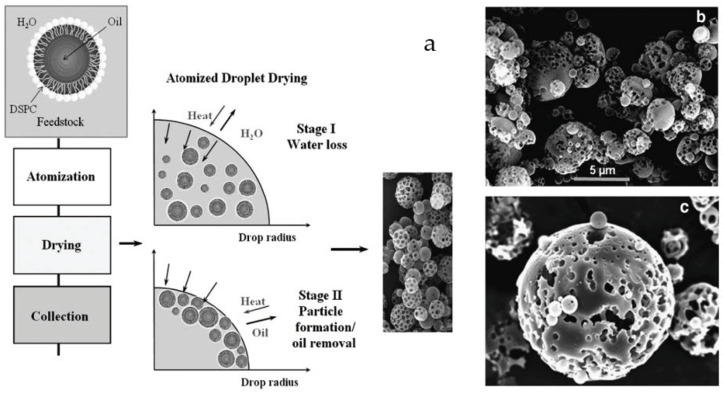
(**a**) Emulsion based spray drying process for manufacturing phospholipid porous particles. Oil-in-water emulsion droplets are created by high-pressure homogenization of perflubron in water. The dispersed oil droplets are stabilized by a monolayer of DSPC. Resulting feedstock is atomized into a hot air stream in a spray dryer. Water is evaporated during the initial stages of the drying process decreasing the particle diameter. Further drying evaporates the perflubron, leaving behind pores in the particle. (**b**) Scanning electron microscope (SEM) images of tobramycin inhalation powder particles and dimensions and (**c**) Tobramycin inhalation powder particle (closeup) [29,87]. Reprinted with permission of Mary Ann Liebert, Inc. Copyright © 2011 Mary Ann Liebert, Inc.

**Table 1 pharmaceutics-14-02495-t001:** Summary of different dry powder inhalers (DPIs) that have been approved or studied.

		Device	Advantage	Disadvantage
Device Type	Capsule	HandiHaler^®^ Single dose High airflow resistance U.S. Approved	Actuated by patient inspiratory flow Used for variety of APIs	Loading the capsule into the inhaler immediately before use may be a maneuver potentially inconvenient for some patients.
		Cyclohaler^®^ Single dose Low airflow resistance U.S. Approved		
		Podhaler^®^ Multi-unit dose Low airflow resistance U.S. Approved		
	Blister	Breo Ellipta^®^ Multi-unit dose Medium airflow resistance U.S. Approved	Intuitive device design	Patients may have difficulty loading and cleaning the blister-based devices. Incomplete pack piercing could lead to incomplete drug delivery.
	Reservoir	Twisthaler^®^ Multi-dose Low airflow resistance U.S. Approved	Intuitive device design Ease of use	Reservoir may not empty with the full dose if the patient holds the device incorrectly.
		TwinCaps^®^ Single-use, multi-unit dose Intermediate airflow resistance Japan Approved		
	Cartridge	Dreamboat Reusable multi-unit dose High airflow resistance Not approved for use	Intuitive device design	Currently not approved for use. Lacks visible or audible feedback that the dose was inhaled correctly as well as a visible verification that the amount needed was inhaled.

**Table 2 pharmaceutics-14-02495-t002:** Asthma classification and treatment in patients ≥12 years of age.

Classification	Symptoms	FEV1 Predicted	Preferred Treatment
Intermittent	Wheeze or cough ≤ 2 days/week, night symptoms ≤ 2 times/month	>80%	SABA PRN
Mild persistent	Wheeze or cough > 2 days/week but not daily, night symptoms 3–4 times/month	>80%	Low-dose ICS
Moderate persistent	Wheeze or cough daily, night symptoms > 1 time/week but not nightly	60–80%	Low-dose ICS + LABA or medium-dose ICS, oral systemic corticosteroids if necessary
Severe persistent	Wheeze or cough throughout the day, night symptoms 7 times/week	<60%	Medium-dose ICS + LABA or high-dose ICS + LABA, oral systemic corticosteroids if necessary

SABA: short-acting beta_2_-agonist, LABA: long-acting beta_2_-agonist, ICS: inhaled corticosteroid, PRN: pro re nata.

**Table 3 pharmaceutics-14-02495-t003:** COPD classification and treatment.

GOLD Stage	Classification	FEV1 Predicted
GOLD 1	Mild	≥80%
GOLD 2	Moderate	50–80%
GOLD 3	Severe	30–50%
GOLD 4	Very severe	<30%

**Table 4 pharmaceutics-14-02495-t004:** Various porous particles generated by different powder technologies for pulmonary drug delivery [80].

**Method of Production**	**Drugs**	**Excipients**	**Outcomes**
Spray drying [91]	Dexamethasone Palmitate (pro-drug of dexamethasone)	1,2-Dipalmitoyl sn- Glycero-3-Phosphocholine (DPPC) and Hyaluronic Acid (HA)	Large porous particles (LPP) containing dexamethasone palmitate; the aerodynamic performance varies depending on the concentration of dexamethasone palmitate, which affects powder cohesion.
Spray drying [92]	Meloxicam	L-leucin, ammonium bicarbonate, sodium hyaluronate	LPP and non-porous particles containing meloxicam for carrier-free formulations were compared at low inspiratory flow rate. The mass median aerodynamic diameter of both formulations was about 2.55 µm, but fine particle fraction (FPF) and emitted fraction (EF) of LPP formulation were much higher than the non-porous counterparts.
Supercritical fluid (supercritical fluid Anti-solvent process, SAS) [93]	Beclomethasone Dipropionate	Poly-ethylene glycol 4000 (PEG 4000). Subcritical water and cold water were employed during the process	The dissolution rate of obtained BDP nanoparticles increases dramatically. The process is called “green” by not using organic solvents.
Supercritical fluid (precipitation of compressed CO_2_ antisolvent, PCA) [94]	Insulin	Poly-L-lactic (PLLA PMs), ammonium bicarbonate	Desired aerodynamic deposition and particle size distribution, and low inflammatory responses because of solvent-free residues. The sustained release pattern provided a similar in vivo hypoglycemic performance to that produced after subcutaneous injection.
Spray freeze drying (SFD) [95]	SiRNA	Mannitol	The integrity of the structure of SiRNA is protected after SFD. The emitted fraction reaches a high value (92.4%), while FPF is unsatisfied (~20%).
SFD [96]	Voriconazole	Mannitol	Optimal FPF obtained with high concentration of voriconazole and tert-butyl alcohol. The dissolution rate of voriconazole was increased.

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
