# Peer review of "Advancements in the Design and Development of Dry Powder Inhalers and Potential Implications for Generic Development"

_pharmaceutics, 2022, doi:10.3390/pharmaceutics14112495_

Round 1
Reviewer 1 Report
The authors showed the different available DPI devices, I suggest collecting them in a table showing the advantages and disadvantages of each one. Also, there is no conclusion in this review. Also, the authors didn't mention anything about the cost of each device.
Author Response
Dear Reviewer,
Thank you for your feedback on our manuscript. Please find attached our response to your queries.
Thank You,
Abhinav Mohan

Reviewer 2 Report
The manuscript by Mohan and coworkers summarizes the current understanding of the formulation and device in DPI performance, past and present research efforts to characterize these performance factors, and the implications that advances in formulation and device design may present for evaluating bioequivalence for generic development. The article is very well written and very interesting I just have few minor comments/suggestions:
- Please when you are talking about the functions of a DPI device, line 112-117, please use bullet points.
- The Figure 5a the resolution is quite bad, please change it.
- In line 550 change “ – “ for “(“
- Regarding the references, you can find the requirements in the section Instructions for Authors. Please, in the text the reference numbers should be placed in square brackets [ ], and placed before the punctuation; for example [1], [1–3] or [1,3]. The references should be described as follows:
·        For  Journal Articles:
1. Author 1, A.B.; Author 2, C.D. Title of the article. Abbreviated Journal Name Year, Volume, page range.
·        Websites:
Title of Site. Available online: URL (accessed on Day Month Year).
Author Response

(The authors gave the same response as above.)
